# Physical Education and Sport between Human Rights, Duties, and Obligations—Observations from Germany

**Michael Fritz Krüger** 

Institut für Sportwissenschaft der WWU Münster, 48149 Münster, Germany; mkrueger@uni-muenster.de

**Abstract:** The starting point entails the declarations of the International Olympic Committee, as well as UNESCO and the Council of Europe on sport as a human right. This article adopts a philosophical and historical perspective on the question of which duties, obligations, and constraints stand in the way of realising this utopian perspective of fair and humane sport as a general human right. The work is based on central historical documents and writings. Two strands of argumentation are pursued. Firstly, the introduction of compulsory physical education, particularly in Germany and on the European continent, in the context of nation-building since the 19th century. Secondly, the idea of a world of sport of its own, which emerged from Olympism and was intended to assert itself against political and economic appropriations. Compulsory physical education is not a human right but a duty. The idea of a world of sports of its own has produced further regulations and obligations in certain fields of sports like professional and commercial sports. Doing sport for health and fitness may become a social obligation.

**Keywords:** human right; Olympism; Olympic education; physical education; fair play

## 1. Introduction

In the charter of the International Olympic Committee (IOC), which the political scientists Rittberger and Boekle also referred to as the "world government of sport" [1], sport is viewed as a human right: "The practice of sport is a human right", is stated in point four of the seven "Fundamental Principles of Olympism". "Every individual must have the possibility of practicing sport", is the specification, "without discrimination of any kind and in the Olympic spirit, which requires mutual understanding with a spirit of friendship, solidarity and fair play" [2].

The IOC thus follows similar formulations in declarations by UNESCO (United Nations Educational, Scientific and Cultural Organization) and the Council of Europe, which are both organisations dealing with cultural developments worldwide and in Europe, including sports and physical education. In Article I of the European Charter "Sport for All" from 1975/76, the following was formulated: "Every individual shall have the right to participate in sport" [3]. In the International Charter for Physical Education and Sport, which was adopted in Paris by the 20th General Conference of UNESCO in 1978, the delegates stated in Article 1 that "the practice of physical education, physical activity and sport is a fundamental right for all". "Physical education, physical activity and sport can yield a wide range of benefits to individuals, communities and society at large" (article two), and, finally (article 11), "physical activity and sport can play an important role in the realisation of development, peace and post-conflict and post-disaster objectives". Therefore, physical education and sport must be guaranteed both in the educational system and in other areas of social life [4].

First of all, these explanations are based on the idea that sport or the practice of sport is free and voluntary. Sport is a human right, not a duty. Secondly, this right is linked to an obligation of the various states and international communities to create the necessary framework and conditions to enable people to realise and exercise their right to sport and

physical education. The Olympic Charter also points out that it is not about any form of sport, but about a sport that is practiced in the "Olympic spirit". Such sport requires mutual understanding—or respect, as Pierre de Coubertin once said—in a spirit of friendship, solidarity, and fair play [5]. In that respect, sports and games are more than and extend beyond individual fun and hedonism, being guided by the spirit of solidarity between individuals, sporting organisations, and states. Considering these universal values and perspectives of sports and games, we should bear in mind that they are based on the traditions of western cultures and civilisations [6].

However, declarations of and on human rights are not descriptions of the reality of life, but ideals intended to guide the actions of people and states in politics, culture, and society. In addition, these mentioned organisations are not powerful in a political sense, but make moral appeals. This also applies to the human right of sport. However, the reality of sport and the practice of sport rarely corresponds to this ideal. How close reality comes to the ideal is difficult to measure; because what individuals understand by sport can be very different and subjective, depending on age, gender, origin, culture, social class, physical fitness and performance, health, etc.[1] [7–9].

Judging by the number of National Olympic Committees recognised by the IOC—currently 206—one can assume that the whole world accepts the human right to sport and that all states in the world strive to implement this right. In reality, however, only a few, rather wealthy states are in a position to provide the corresponding infrastructure for sport and physical education. Cultural obstacles and ideologies often prevent the participation of women in sports, and lead to racism or discrimination of persons with handicaps. The material and organisational or institutional obstacles are only a kind of "luxury problem" on the way to implementing the human right of sport. More serious are probably the political and social conditions in numerous countries which, although they are recognised member organisations in the IOC, nevertheless restrict general freedom and human rights in such a way that free and voluntary sport in the Olympic spirit is either not or barely possible for the individual.

Finally, sport as a social phenomenon of modernity has produced a plethora of duties and obligations by both individuals and the state, constraints and coercive structures that certainly limit the freedom of the individual to develop his or her "physical, mental and moral powers", as programmatically formulated in the UNESCO Declaration of 1978. These constraints are manifold, ranging from rigid forms of physical education or drill in and through sport, especially in schools and the military, to the complex constraints of high-performance sport, professional and commercialised sports, the power of media in sports, extending to the compulsion to be sporty in order to conform to an ideal of the modern, dynamic and fit, high-performing human being. These structures including both their formal rules, duties, and obligations plus rather informal constraints of the system(s) of modern sports are widely analysed and discussed in current sociological research [10]. The German context is considered by the work of Karl-Heinrich Bette claiming that the genesis of sport as a social phenomena (and "system") is both an answer or reaction of the growing regulations and contraints of modern societies for individual freedom but simultaneously producing new, sporting rules and constraints according to social structures and regulations [9–11].

The term constraint in this context is based on the thesis of the sociologist and "human scientist" (*Menschenwissenschaftler*) Norbert Elias, author of the paradigmatic *oevre* of the "process of civilisation". This process is characterised by the fact that people are exposed to constraints that take on different forms in the process of social development, depending on the social figuration. The process of state- and nation-building means that individuals are forced to exercise greater control over themselves. This process can also be observed in sports development, as Elias and Dunning (1986) have demonstrated [12,13]. Elias essentially distinguishes between constraints that emanate from nature, and other social constraints that people exert on each other interdependently in certain social contexts or figurations. In each case, such constraints limit the freedom of the individual. Technology

and civilisation have contributed to the fact that natural constraints which cannot be directly influenced by humans seem to be more controllable, for example natural events or catastrophes, or overcoming great distances through modern means of transport and communication. Currently, in facing climate change, humankind has to realise that the power of nature strikes back, caused by humans who intend to control and manage such constraints emanating from nature.

In addition, social constraints have definitely tended to increase. For Elias, the process of civilisation is characterised by a shift in social constraints from external constraints to more self-coercion and self-control. People learn to control themselves more. Elias speaks of "social compulsion to self-compulsion"[2] [12]. At the same time, they limit their freedom of action by being considerate to others. Both can be observed in sport; on the one hand, sport offers people numerous opportunities to overcome constraints and thus break through limitations on behaviour and individual freedom. On the other hand, sport itself is an area of life in which new, additional rules, norms, and thus restrictions on individual freedom are invented and enforced [12]. Sport generates and overcomes social constraints in equal measure.

My perspective in this paper is that of considering the genesis of physical education, exercises, and sport, located between ideal and reality subject to the question of how movement, physical exercises, games, and sport became a symbol of hope for freedom and human rights on the one hand, and how new constraints and dependencies developed in and through sport itself on the other hand, shatter hope again and again.

Considering the genesis of physical education, exercises and sport as a "human right", we should keep in mind that this claim is a product of modern society. Sport was originally the exclusive privilege of the wealthy classes of Western Civilisation, mostly for its younger male members. Physical education was part of an educational concept of the middle classes, first reserved for young white men. "Sport for all" marks the temporary status of a process of sportisation of leisure for the masses, which was and remains a privilege of the wealthy who can afford to do sports, because they have enough time and money.

The realisation of the human right of sports for all is in fact a vision and less a reality for most people in the world, who suffer from poverty and malnutrition. Without a minimum material standard of living, i.e., sufficient food, clothing, housing, work, and income, it is barely possible for individuals to use their human right to sport and exercise. From their perspective, the claim of sports as a human right may even be greeted with cynicism.

This is what Willi Daume, former Vice-president of the IOC and president of the Organizing Committee of the Olympic Games in Munich 1972, meant when referring to the "Democratization of Sports" at the conference of the International Council of sports and physical education in Mexico in 1968. "The world is in distress", he stated to the international audience, and continued: "And we want the whole world to play. Do we have the right to wish for such a thing?", he finally asked. As expected, he gave a positive answer and reiterated the human right to sport, for all sexes, ages, and social classes, especially for "minorities" and "underprivileged". For them, sport could contribute to "make life more liveable", he stated, and "support vital forces" [14].

## 2. Ernst Bloch and the Hope for Freedom and Human Rights through Physical Exercises and Education, Gymnastics, and Sport

Ernst Bloch (1885–1977) is called "the philosopher of hope" because his main work "The Principle of Hope" was a major success and translated into many foreign languages. Bloch wrote it in the USA in exile during the Nazi period (between 1938 and 1944) when hope seemed to be more unattainable than ever. In this book, the Leipzig Marxist also mentioned German gymnastics (*Turnen*) and sport in two places in, but without considering the subjects in depth. "The young gymnast thought of freedom", he wrote about the beginnings of Friedrich Ludwig Jahn's concept of German gymnastics over 200 years ago. "This freedom, however: walking upright, strength, not cowering before the enemy

but standing one's ground, male pride before royal thrones, civil courage, did not come afterwards, as is well known"[3] [15].

Bloch wrote about sport: "Sporting exercise also remains a wishing, hoping exercise. It also wants to do more with the body, to be able to be more than was sung to it in the cradle." However, he sees this utopia of sport disappointed by social conditions, the power of circumstances. "Only in an unsuppressed people can hope become reality. Sport needs freedom" [15].

With these two examples, Bloch mentioned two different types of utopias that are still associated with sport today and have been incorporated into the idea of a human right to sport. One refers to the utopia of being able to achieve political and social freedom and self-determination in and through gymnastics (*Leibesübungen*) and sport. On the one hand, gymnastics stands for the specific development of body exercises in politics, culture, and society in Germany, but on the other hand, it is also a metaphor for physical exercises and physical education as practised everywhere and referred to in English as body exercises, physical exercises, or drill.

The other form of utopia indicated by Bloch relates to the physicalness of man and the limits imposed by the nature of his body. Sport is associated with the illusion of being able to transcend these limits, to make more of him than was "sung to him in the cradle".

Both utopias and their disenchantment will be examined in more detail below. Both also represent different constraints under which sport takes place, on the one hand external, political, social, and cultural constraints (1), and on the other hand constraints or limits imposed on us as human beings by the fact that we have a body and have to (learn to) deal with it—in different ways, one of which is represented by the institution of sports and games (2).

### 3. German Gymnastics (Turnen und Leibesübungen) and Physical Education (Leibeserziehung)—Social Utopia and Political Constraints

The first example refers to the political hopes associated with the Turner movement (gymnastics movement) in Germany. Friedrich Ludwig Jahn (1778–1852), with whom the founding of the gymnastics field on Hasenheide in Berlin in 1811, and thus the beginning of the German (and international) gymnastics movement is linked, was concerned with freedom through gymnastic exercises, play, and games. Physical exercises with and without gymnastics equipment as well as games, which in sum he called "Turnen"—a term which seems to be untranslatable—were to be used to gain or fight for social and political freedom. He and his young followers took the right and freedom to do gymnastics, Turnen (i.e., gymnastics) 200 years ago was what sport means today, basically the totality of all organised physical exercises, play, and games. They were concluded with a clear political intention. Their aim was to do gymnastics freely and simultaneously, as a means of fighting for the freedom of the fatherland. Specifically, they meant firstly, liberation from the yoke of the French occupiers, namely, the Emperor Napoleon. Secondly, they associated this liberation from the external enemy with the hope of a German nation state in which the citizens themselves could determine their fate. They opposed the "royal thrones", as Bloch put it, and fought for the freedom of the people and the fatherland. "To do vigorous gymnastics for people and fatherland" was a slogan of Jahn and the early gymnastics movement[4] [16–20]. For them, German gymnastics was a source for realising a political hope.

### 4. Jahn and the Early Gymnastics Movement

In addition to the "cause of the fatherland", gymnasts in Germany were also concerned with a better and more consistent education of the youth in body and mind. Politics and pedagogy, nation and gymnastics belonged together. Jahn did not himself use the term nation, because it was a foreign word, and he was against everything that he considered "foreign" and thus "un-German". At that time, Jahn and his patriotic followers included only boys and young men, not girls and women. Today, Jahns' attitude might be regarded as racist, when "foreigners" and foreign languages were excluded from German Gymnastics,

and sexist, when only young men and boys should do gymnastics but not women and girls. Although these restrictions and limitations of the general "human right" for sports and gymnastics were reasoned by the contexts of ruling values of culture and society in the period of nationalism, there is no doubt that at the beginning of the German gymnastic movement, the hopeful and optimistic vision of physical education and sport for all was limited by sexist and racist restrictions.

Among others, the exclusion of girls was one of the differences between German *Turnen* and educational gymnastics as taught and propagated by the philanthropist Johann Christoph Friedrich GutsMuths (1759–1839) and the educationalist Johann Heinrich Pestalozzi (1746–1827). In the view of Jahn and his followers, *Turnen* was more than simply gymnastics based on the model of antiquity, but "patriotic" gymnastics, part of a popular "customary art", as Jahn expressed it in the book "Deutsche Turnkunst" (German Art of Gymnastics), which had been lost due to the fragmentation of Germany, as he propagated [16] He claimed to have revived it, and to do so, he said, it was first necessary to free Prussia and Germany from the domination of foreign powers, especially France and Napoleon.

That one had to free oneself from this yoke of French foreign rule, was the political and at the same time military message of the gymnastics father. Clearly, Jahn was not a pacifist, but one could say in today's words, he felt as and was a kind of freedom fighter, like Andreas Hofer in Tyrol, who was also not afraid to fight for the freedom of the people and the unity of the nation by means of violence. However, there is a fine line between a terrorist and freedom fighter. Ultimately, the games and exercises of the young, male gymnasts on the gymnastics courts and later halls in Germany also served to prepare them not only ideologically but also militarily for the fight for freedom. Nevertheless, *Hasenheide* was not a military training camp. Weapons training was not systematically practised, even if "Gere" (scissors) were used, which were thrown far and wide into the target as a gymnastic exercise—along with other gymnastic exercises on the parallel bars, high bar, and the swing. The *Hasenheide* and other gymnastics courts and halls in Germany remained educational spaces, training grounds where, in addition to physical fitness, patriotic attitudes were to be acquired.

Gymnastics in Jahn's time was a highly political, and at the same time educational matter. Paradoxically, the Turners' fighting for a free fatherland was achieved by discipline and even aggression and bodily power against their enemies. Nevertheless, the meaning of what is political and what is pedagogical has changed considerably. This change in meaning already began in the early 19th century. When Napoleon was defeated and Prussia had taken the lead in the national movement, the "cause of the fatherland" was no longer considered political, but a "natural" attitude of every citizen. However, what exactly defined the German fatherland, as Jahn's friend Ernst Moritz Arndt (1769–1860) had commented in his poem, was not clear for a long time[5]. Since many gymnasts adhered to the option of a large Germany including Austria, and furthermore favoured a German nation state in which state, politics, and society would be regulated and legitimised by a constitution, concluded by representatives of the people, they were seen as politically opposed to the ruling regimes. This was particularly evident in the German, in fact European Revolution of 1848/49, when many gymnasts of the kingdom of Swabia and the dukedom of Baden went to the barricades and even fought for a free German republic. This was in stark contrast to their gymnastics father Jahn, who held up the flag of the monarchy as a deputy in the Paulskirche in Frankfurt where the first constitution of a democratic German nation state had been discussed and concluded. Regrettably, this free, liberal, and democratic constitution never became reality, but remained an ideal for further democratic constitutions of the Weimar Republic (1919) and the Federal Republic of Germany (1949). For this reason, because he was against the republic, Jahn was almost lynched, as indeed happened to his parliamentary colleagues General Hans von Auerswald and Prince von Lichnowski, in whose murder radical gymnasts are also said to have participated. Jahn himself just managed to escape the angry crowd [19].

## 5. Exercises and Drills

At the same time, the politicians of the developing nation states quickly recognised the benefits to the state and nation of systematic physical training and education of subjects and citizens. Gymnastics, drills, and body exercises gradually became an integral part of the education of young men in school and the army, and gradually of girls and young women as well. Thus, a principle of the early gymnastics movement was abandoned, which Jahn himself had formulated in his book "Deutsche Turnkunst": "The gymnastic place is not a drill place, and therefore cannot be stuffy like at school" [16]. School gymnastics, school sport, and military sport in the army were and are still not free, voluntary, and deliberate actions of the people, but are compulsory within the education system defined by the state, nation, and society. *Leibesübungen*, *Turnen* (Gymnastics), games, and sport as physical education are part of bio-power, said Michel Foucault[6] [21]. As part of the nation-states and its power, physical education became a means of (self-)disciplining and controlling the bodies and minds of young members of the society.

Adolf Spieß (1810–1858), who is regarded as the founder of physical education at schools, wrote a comprehensive didactics and methodology for school gymnastics, body exercises, and physical education including educational instructions. However, he did not consider the systematic practice and learning of physical or gymnastic exercises as a contradiction to Jahn's dictum that the gymnastics arena should not be a place of drill. His argument, which was shared by the majority of PE teachers of the 19th century, was rather that the individual was only truly free when he could control his body (and thus himself). Therefore, it was necessary to train (or exercise) the body in order to be able to control and use it as an effective tool: "We want to train the body as the spirit's freest tool, at the same time with it and for it, so that our life, out of its predominantly mental withdrawal, may settle more at home in the body, that it may become less floating and blurred, and instead grow and flourish more firmly, healthily and completely" [22]. A lack of body and self-control was seen as an expression of the greatest lack of freedom.

The gymnastics clubs that were founded before the revolution of 1848/49, in the pre-March period of the 1840s and in the revolutionary period around 1848 actually had nothing to do with Jahn. Their focus was on southern Germany. As gymnasts, they were primarily freedom fighters. They organised themselves formally in clubs but supported the introduction of school gymnastics in the hope of a liberal and democratically constituted nation state. In Esslingen, they joined together on 1 May 1848 to form the Swabian Gymnastics Federation (Schwäbischer Turnerbund). This association was the nucleus for the foundation of a national umbrella organisation for physical education, gymnastics, and games under the banner of "Deutsche Turnerschaft". The Swabian Gymnastics Federation still exists today. However, the German Gymnastics Federation was dissolved by the National Socialists in 1935, as were the workers' gymnastics and sports organisations. The German Gymnastics Federation and the German Sports Federation took over their legacy after 1945 [17,20].

## 6. Dashed Hopes

The idea of the political gymnast, who exercises body and mind, and fights for the freedom of the people and the fatherland, weapon in hand, was already buried after the suppression of the 1848 revolution by the Prussian Crown and especially the "Kartätschen-prinz" ("catridge-prince") Friedrich Wilhelm IV. Finally, the German Gymnastics Association increasingly professed a kind of educational gymnastics or physical education in which the education of the body had to be in the foreground. Political and military ambitions were no longer pursued. Instead, a new understanding of gymnastics and politics developed in the clubs of the German Gymnastics Federation. The gymnasts were intended to and indeed wanted to get involved locally in the towns and communities for the common good, for example in the fire brigades and ambulance service. They thus contributed to creating a basis for civic, democratic engagement in Germany, first in the German Confederation, then in the German Reich, in the Weimar Republic and finally in the Federal Republic of

Germany. Except for the Nazi period, the common sense relating to the concept of sport and gymnastics was the independency from state politics. Sport and gymnastics should be an individual right. The state should support this right but not dominate free movement and sports. However, the nation-state ought to be taken the responsibility and obligation for physical education at schools of every child. In sum, the German sports and gymnastics organisations contributed to the process of cultural nation-building of the Germans [17,20].

However, the Nazi period must not be concealed in this context. Many gymnasts and athletes now denied their democratic and civic traditions. They wanted to march into the Third Reich "side by side with SA and Stahlhelm", as the "Turnerjugendführer" (gymnastics youth leader) Edmund Neuendorff (1875–1961) had formulated it, without, however, asking the gymnasts and also SA and Stahlhelm. The latter did not want the gymnasts at their side at all ([23], p. 166). In principle, the same applied to the entire bourgeois sports movement. Apart from the gymnasts of the German Gymnastics Federation (DT), this also included the large sports associations, the German Football Association (DFB), as well as the track and field athletes, swimmers, handball players, skiers, and many others. They adapted to the political system or submitted to the constraints and controls exercised by the state and the party. Gymnastics and sports organisations like the socialist and communist workers sports and gymnasts associations that still stood in political or ideological opposition to the regime were banned and persecuted [23].

Gymnastics and sport in Germany did not succeed in implementing their utopia of political freedom through sport. Instead, they ended up in a dictatorship. Gymnastics and sport did not serve as a means achieving political and social rights and freedoms, and the opposite was the case. This is probably what Ernst Bloch, who has already been quoted, had in mind when he stated: "Physical exercise, without that of the head, meant after all: being cannon fodder and before that a thug" [15]. He argued that physical exercise and sport need to be associated with an awareness of humanistic values. Without this spirit, physical exercises may be used barbarically.

Moreover, the political framework conditions of a dictatorship did not allow gymnastics and sport to be realised as an independent, autonomous area of a "world of their own" in which freedom, courage, and civil courage could be experienced and lived independently of politics and society in a peaceful world of sport. This did not succeed in the second German dictatorship either, namely, the rule of the communist party SED, essentially the regime in the East-German Democratic Republic (GDR). Physical culture and sport were not free and independent, but subordinated to the directives of politics, up to and including state-imposed doping. However, there was room for manoeuvre or "niches" that were also used by those people living in the GDR [24].

## 7. The Eigenwelt of Sport—A World of Its Own

The model of sport as a world of its own existing beyond politics and society, was Pierre de Coubertin's idea. In numerous writings, the founder and spiritual father of the modern Olympic Games formulated the idea of a sporting and athletic geography of its own, independent of politics and state. Sport, according to Coubertin, established a world of its own [25]. In Germany, Carl Diem (1882–1962) attempted to anchor this ideal of an ideal sports world, independent of politics and economics [26]. With the Olympic Games, a separate, worldwide model of international sport was constituted with the aim of using sport to set an example of people's ability to perform and progress, as well as of peaceful and respectful interaction with one another in sport and fair competition. Seen in this light, the Olympic Games and the philosophy associated with them stand for an attempt to make the world a better and more peaceful place with the help of sport. Everyone should be able to participate in sport, without discrimination on age, origin ("race") or gender, according to the rules and norms of sport, peacefully and fairly, as indicated in the Olympic Charter Only the sporting performance of the athletes should be decisive for their selection. In line with this ideal, the IOC sees itself as an advocate of a worldwide human right to sport, independent of states and governments.

In reality, however, the implementation of such ideals has always left much to be desired. In the history of the Olympic Games in modern times, attempts have been made again and again to uphold this Olympic utopia in the face of political and economic influences. However, the realisation of this utopia of making the world a more peaceful and fair place through peaceful and fair Olympic sport is still a long way off. "I am always amazed when I hear people saying that sport creates goodwill between the nations," wrote George Orwell [27] in the London Tribune newspaper in 1945, "and that if only the common peoples of the world could meet one another at football or cricket, they would have no inclination to meet on the battlefield. Even if one didn't know from specific examples (the 1936 Olympic Games, for instance) that international sporting contests often lead to orgies of hatred, one could deduce it from general principles."[7]

## 8. Utopias of the Athletic Body and Its Real Constraints

How successful was and is the hope mentioned at the beginning by Ernst Bloch for a sport that promises to make more out of the body (and mind) than was sung to it in the cradle? This utopia has two essential dimensions. The first involves the idea that competitive and high-performance sport is able to implement a special model of increasing physical effort and performance. *Citius-altius-fortius* is not by chance the motto of the Olympic Games [25]. Coubertin explained it in 1901 as follows: "In this way [i.e., the sportsman, author] is able to cultivate effort for effort's sake, to seek out obstacles, to place a few obstacles in his own path, and always to aim a little higher than the level he must achieve." The Olympic motto corresponds with the promise of progress in the modern, industrialised, capitalist world—but without utilitarian thinking, as Coubertin explicitly emphasised: "The sportsman remains a stranger to utilitarian concerns. In fact, sport generates record physical performances." Coubertin saw this as both an evil of modern sport and its poetry (ibid.)[8]. He resolutely opposed imposing restrictions on the need to want to be better. "Ses adeptes ont besoin de la 'liberté d'excès'," he commented in his French mother tongue on the Olympic motto in his famous radio address in 1935, a year before the 1936 Olympic Games, saying it applied to all those who "osent prétendre à abattre les records!" [28]. In Olympic high-performance and elite sport, the athlete pushes the limits of physical capacity, but in Coubertin's sense also to the moral limits—and sometimes beyond.

The second dimension of the body utopia of sport consists of its promise of naturalness and health. Included in this is the idea of being able to become or remain physically (and mentally and spiritually) healthy and "natural" in and through sport, but also free, attractive, young, and capable. Freedom in this context, also and above all, means being free from physical impairments and infirmities. Movement and sport are regarded as a general remedy to promote better health, well-being, and long-lasting youthfulness.

It is no coincidence that sport thrives on images showing young, muscular, athletic, outwardly healthy, and mostly confidently smiling people. Athletes often present their bodies in light clothing, i.e., in a "natural" state. The unclothed, natural body has often been considered an expression of freedom, impartiality, and emancipation from civilisational constraints. The social philosopher Hermann Lübbe speaks of "body emancipation" and in this context, tells the story of Adelheid Amalie Fürstin von Gallitzin, who lived in Angelmodde near Münster from 1779 and let her children bath naked in the small river Werse. Goethe reports in an admiring tone about the countess, who belonged to the circle around the enlightened reformer Franz Freiherr von Fürstenberg (1737–1825), chief official of the prince-bishopric of Münster and an important representative of the Catholic Enlightenment. "Laced chest and heel disappeared, the powder dissipated, the hair fell in natural curls," Goethe says of the obviously attractive countess. "Her children learned to swim and run, and perhaps also to scull and wrestle" [29].

### 9. Physical Emancipation

According to Lübbe, "body emancipation" in the sense of a "recovery of the ideals of naturalness" has been a fundamental motif of body exercises, gymnastics, games, and sport since the European Enlightenment. However, the "gain in freedom in relation to one's own body" was also accompanied by greater control and discipline of the body. Care of the body becomes a norm, a duty, a pedagogical responsibility, a health task, and so on. According to Lübbe, this dilemma still shapes civilised health sport and modern body cult today. With the development of the mass movement, physical emancipation became a political factor. Lübbe comments that the intertwining of sport and politics has many faces, ranging "from nationalism to totalitarianism and from the hope for world peace that was and is associated with the Olympic idea to the socio-political expectations of financial relief for the social coffers through health-promoting mass sport" [29].

Only a few athletes can become Olympic ones, but most people want to be sporty, fit, powerful, healthy, and attractive, and to be seen as such. They see sport and exercise as a way to achieve this goal, which is both social and personal. There is a powerful health or health sport industry that tries to meet the need described by Bloch to make more out of the body than is sung to it at the cradle; for from birth it is certain that our bodies degenerate and that we must die in the end. Through sport, people believe they can counteract or at least slow down this decay. Through sport and exercise "stay 40 for 20 years" was a motto of the German sports physician Wildor Hollmann, which has meanwhile become a commonplace in popular and health sports and preventive medicine [30]. Looking around today, the motto could be: "60 is the new 40".

The need for health, performance, and fitness in and through exercise, gymnastics, gymnastics, games, and sports are the main reason for the mass spread of sport in modern society. The term "sport" in this context also includes the systematic exercise and training of physical performance factors, thus encompassing all types of physical training referred to in earlier times as physical exercises, physical education, gymnastics, or "drill". As a mass phenomenon, sport is part of the public health movement, to which not only sports organisations but also school sports, commercial sports and fitness studios, health insurance companies, family education centres, and other preventive and rehabilitative institutions in the health sector are committed.

### 10. Tendency towards Excess

Contrary to the ideal that the founder of Olympism had imagined, the tendency of Olympic high-performance sport towards "excess", as recognised by Coubertin and quoted above, is nowadays no longer based on an autonomous, free will decision of the individual to "want to break records". High-performance athletes are in fact under great (and probably ever-increasing) external constraints imposed by politics and society. After more than 100 years of competitive sports development, top sporting performance can no longer be achieved as an individual athlete—Coubertin called this type of male individual athlete "désbrouillard"—and now only as team performance[9] [31]. The individual athlete functions only as part of a system that has the task of producing top sporting performances. This team or system includes trainers and coaches, masseurs, and physiotherapists, as well as nutritionists, career advisors, advisors on dealing with the media and taxes, and numerous other experts. They are all there to help guide the athlete to peak performance. As part of the system, the athlete not only has to submit to strict rules and controls during training and competition, for example, sometimes also to degrading doping controls that invade the private sphere, but he or she is also forced to completely subordinate his or her entire lifestyle, family life and life planning to this overarching goal of top sporting and Olympic performance. To speak as did Foucault, professional elite sport has established its own bio-power, which has partly taken on quasi-totalitarian features comparable to previously known forms of bio-politics in state, economy, and society. The Finnish sports scientist and sociologist Kalevi Heinilä recognised this development as early as the 1970s and described it as a process of the "totalisation of sport" [32].

## 11. Resumée

### 11.1. Conclusions

Sport is more than a "natural" human right to free movement. It is a phenomenon of modern society and an expression of its culture including their various regulations, duties, and obligations. As such, it is also subject to the manifold constraints from politics, culture, and society. Free sport or the exercise of the human right to movement, play, and sport can therefore only be realised within these contexts. A global and national sports policy intended to provide a human right to sport is therefore obliged to strive for a framework of conditions in politics, the economy, and society that enable individuals and athletes to engage sport in an atmosphere of freedom and self-determination.

We analysed two mainstreams of the development leading to the vision of sports for all as a "human right", with respect to the German context in particular. One stream entailed the connection of sport and physical education to the educational concept of the middle classes. Physical education and gymnastics were and still are a means for educating or enabling people to use their right to move freely and independently. Paradoxically, one needs discipline and education to work for this right. School sport and physical education are both a human right and a formal duty. Pupils have to take part in physical education, organised and sanctioned by state schools. In this educational sense, the path is described in the Latin saying of "per aspera ad astra"—through aspiration to the stars, achieving one's goals through effort.

The other was the process of democratisation, extending the privilege of sport for the wealthy to all, in principle. Elias called it "functional democratization" [33]. Marginalised groups, especially women and older people, gained access to sports—"for all" [34]. A result was the concept of sports as a world of its own, accessible to everybody longing for a better life, including both health, vitality, and performance. Sport for all and the human right for sport are two sides of the same coin. Both mainstreams of this development for the vision of sport as a human right were closely connected to the political, social, and cultural contexts. These structures may also stand in the way of realising a human right to sport, when the right turns to its opposite, duty and compulsion. If athletes are forced into sport through political and economic imperatives, this is not compatible with a human right to sport in peace and freedom. Totalitarian politics and dictatorship prevent free sport just as much as the dominance of economic exploitation.

### 11.2. Consequences and Limitations

As paradoxical as it sounds, it seems that high-performance athletes are also denied the exercise of the human right to free sport. While no one is forced into high-performance sport, those who choose this career path must be prepared to accept numerous restrictions on their freedom. For example, they are effectively forced to play sport every day, pushing the limits of their performance, if they want to participate in international competitions and tournaments at the highest level. Training, competition, and controls in high-performance sport sometimes massively restrict the freedom and even human rights of athletes. This applies not only to the sometimes harsh training methods, but also to the—at least in some countries—strict doping controls. All athletes must submit to such procedures if they want to participate in high-performance sport. The fight against doping has led to athletes having to voluntarily accept intrusions into their private and intimate sphere that are hardly compatible with constitutional methods and the principles of data protection.

Finally, cultural and religious factors also stand in the way of realising the human right to free sport. If girls and women are prevented from practising sports by strict dress codes or gender-role stereotypes, such measures also restrict the human right to sport.

Since sport in the modern world is not only a social phenomenon, but also a cultural achievement that needs to be nurtured and developed, it is necessary to work constantly on a free, humane, and authentic culture of sport, as sports educator Ommo Grupe (1930–2015), one of the founders of modern sport pedagogy in Germany and beyond, wrote. This goal of a demanding, humane everyday culture of sport cannot prevail "if competition and rivalry

dominate even where partnership is appropriate; if the less able are discriminated against compared to the more able; if advertising and marketing take uncontrolled possession of sport; if young children are already trained for top performance; if access to sport remains barred to anxious or disabled people or foreigners. Or when in competitive and top-class sport, only performance, and success count and not the athletes who achieve them; when they have to provide extras for the appearances of the big stars; when nationalism displaces expert judgement; when glamour and happenings are passed off as performance, or competitive sport degenerates in a media spectacle, or politics and business dictate its goals." [35].

. . . -then it is time to reinvent sport as a humane human right.

Seen in this light, the "sporting exercise ( . . . ) remains a desiring, hoping one", as Ernst Bloch so aptly put it in the Principle of Hope: "It also wants to do more with the body (and mind), to be able to be more than was sung to it in the cradle."

**Funding:** This research received no external funding.

**Data Availability Statement:** Not applicable.

**Conflicts of Interest:** The authors declare no conflict of interest.

## Notes

[1] The difficulties of defining sport are repeatedly pointed out in the literature. See, for example, Krüger, "Sport–Begriff und Geschichte" [7] as well as in the international context Fry, "Sport" as well as Simon, "Theories of Sport" [8,9].

[2] See especially the remarks in the concluding chapter "Draft for a Theory of Civilisation", Elias, pp. 312–454, here pp. 312–335 [12].

[3] Bloch, *Das Prinzip Hoffnung* Volume 2, p. 524. The book was written in exile in the US between 1938 and 1944. The title was originally intended to be "The Dreams of a better Life" [15].

[4] For a broader overview and contextualization of the history of German gymnastics, physical education and sport since the 19th and 20th century see Krüger [17,18,20].

[5] "What is the German Fatherland?" was the famous song by the romantic-nationalist poet Ernst Moritz Arndt (1769–1860). He wrote it the lyrics in 1813, the year of the Battle of the Nations, and Gustav Reichardt composed it in 1825. https://de.wikipedia.org/wiki/Des_Deutschen_Vaterland (accessed on 18 June 2016).

[6] See the research by Reinhart, *"Wir wollten einfach unser Ding machen"* on the politics of sport in the GDR in the light of Michel Foucault's philosophy [24]. Foucault himself also referred to the pedagogy of philanthropists as a particular "technology of power", especially in his work Foucault, Raulff, and Seitter, *Der Wille zum Wissen* [21].

[7] Orwell wrote this sentence in 1945 in an article entitled "The Sporting Spirit", given his impression of both the end of the Second World War and the beginning of the Cold War and the memory of the 1936 Olympic Games in Berlin, which on the one hand had celebrated the idea of peace in Olympic sport, but on the other hand had marked the beginning of an era of tyranny and war. See Orwell and Packer, *Facing Unpleasant Facts* [27].

[8] The article "La psychologie du sport" appeared for the first time and in French in the journal "La Revue des Deux Mondes", 70e année, 4e période, tome 160, 1er juillet 1900, pp. 167–179.

[9] See Coubertin's speech on 30 June 1907 at the Sorbonne in Paris, in Coubertin, *Einundzwanzig Jahre Sportkampagne*, pp. 169–173 [32].

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
