# Peer review of "Physical Education and Sport between Human Rights, Duties, and Obligations—Observations from Germany"

_societies, doi:10.3390/soc11040127_

Round 1
Reviewer 1 Report
The paper makes an overview of the development of the concept of sport in the modern societies and particularly focuses on the principles of its constitution as a human right. It also draws on the development of the practice of sport and physical activity in the historical German context. The general theoretical framework of the paper is based on Elias’ concept of the process of civilization and the way sports become an activity related to self-control and care for body and health. In this regard my suggestion is to add a paragraph on the constitution of sport as an activity related to leisure time as a distinct sphere of modern society opposed to the sphere of work. In the current version of the paper this line of the analysis appears at the end but it seems to me that it is very important and can be added in the beginning of the section on the historical reconstruction of the practice of sport. The paper emphasizes more on the constriction of sport as a human right but it might be useful also to reflect more on the process of professionalization of sport and its distinction from sport as a leisure activity. A third line of the analysis that can be added relates to the changing concept of health and care for body which in modern society becomes a central value and an object of personal proactive behavior. What might also be interesting to reflect on in the paper is the process of commercialization of sport as part of the flourishing industry of wellness in the modern ‘youth-driven’ culture. These are different lines of theorizing sports that can be added to the main concept of the paper on sport as a human right.
Author Response
Thank you for your valuable suggestions on revising the manuscript.
I will endeavor to take your mentioned points into account in the revision.
Please see the revised manuscript.
Reviewer 2 Report
(Please note major comments in bold.)
Abstract
Include the key findings: which were the most significant constrains the authors identified?
Main Text
line 16: I think the authors could engage more with existing scholarship on topics like physical education and sport as a human right, for example. The authors did a very good job here with drawing on primary sources, but it is not as clear what scholars have previously contributed to the topic of how constraints have limited the notion and practice of sport as human right. Explaining this might help the authors do make more clear how the present work adds to our understanding of these phenomena.
line 24: UNESCO. The first time you use an abbreviation, spell out the full name or title and introduce the abbreviation in parentheses. After that use only the abbreviation.
25: Explain the background of the European Charter. Provide some context.
line 33: “Therefore, physical 33 education and sport must be guaranteed …” Though it is laudable that the several organizations have taken this stance, their jurisdiction and power to enforce those guidelines and/or policies are limited. The IOC, for example, can only create and reinforce policies within their own organization and events. Thus, “must be guaranteed” seems to be a hopeful statement. In fact, the limits of enforcement could be considered a major constraint. As you noted in lines 49-50, these declarations are “ideals intended to guide the actions of people and states in politics, culture and society,” not national or international policies.
line 56-57: “one can assume that the whole world accepts the human right to sport and 56 that all states in the world strive to implement this right” In my opinion, this interpretation is a stretch. Some member NOCs, for example, might reluctantly accept this ideal of the IOC, but be fare from striving to implement it. Based on the following sentences, it appears that you agree with my assessment, thus consider changing “In fact…” to “in reality, however,…”
line 72: Consider adding a citation at the end of the paragraph, pointing to support for this assessment.
line 89: Add page numbers for the direct quotation. For embedded citations in the text with pagination, use both parentheses and brackets to indicate the reference number and page numbers; for example [5] (p. 10). or [6] (pp. 101–105). Correct this throughout the document. I will not point this out again after this, so you will have to check on your own.
line 96: “This paper is about looking at..” (a) Avoid anthropomorphism ("the attribution of human characteristics or behavior to a god, animal, or object”). For example, instead of writing "the textbook discussed," refer to the author(s) instead. If the editors of this journal agree, use first-person. (Free yourself of the constraint to have to appear impartial.) Also, phrases like "...looking at..." sound very casual. How about "we examined" or “we discuss”?
line 103: “The philosopher of hope…” Explain this phrase more.
line 107: “…over 200 years ago” It might not be quite clear what event occurred over two hundred years ago. It could be the rise of the Turner movement or Bloch’s writing.
line 110: “Bloch writes…” Use past tense when you discuss “past” literature.
line 113: What did Bloch mean by “unprinted people”?
line 119: “gymnastics stands for…” Do you think it still stands for these ideals today? Did you mean to say that it stood for this originally?
line 123ff: Avoid short paragraphs like this.
line 132: “3. Re 1)” It is not clear what this means in the header.
line 142: “claiming their human right to sport” Did Jahn and/or others in the early Turner movement refer to Turnen as a human right?
line 151: I think the footnote text for footnote 4 is incomplete (see lines 479-480.)
line 155: “a better and more consistent education of the youth in body and mind”
Where the Turner equally concerned about this education for girls and boys, women and men? If not, “youth” might be too broad here. On a related, more general note: Regarding Jahn and the early Turner movement, consider discussing sexism and anti-Semitism, as these relate also to the constrains limiting the hopeful vision of physical education and sport.
line 161-162: I know what you mean here, but this phrase could confuse readers: “which is the English equivalent of the (German) gymnastics.”
line 172: “like Andreas Hofer in Tyrol” Many readers will need more context here.
line 182: Guide the readers’ understanding more here. Remind them how all of this ties back to the discussion of sport as human right and its constraints. Currently, this section could be written for any number of papers about the Turner movement. Make the discussion more specific to the main topic of the manuscript (“Physical Education and Sport Between Human Rights, Liberty and Social Constraints.”) Connect the information in these sections to Elias’ theoretical framework. In my opinion, this critique is an important one and one to consider for other parts of the manuscript as well.
line 202: “they were seen as politically oppositional” By whom? Use active voice.
line 207-208: “Regrettably” Explain why this was regrettable.
line 225: “are part of bio-power, said Michel Foucault” You make a good point here, but explain this more.
line 232: Insert a citation at the end of this sentence.
line 237-238: “A lack of body and self- control was seen as an expression of the greatest lack of freedom.” In my opinion, there is a great opportunity here to connect this discussion to the main topic of the manuscript. Explain more how all of this is related to “Physical Education and Sport Between Human Rights, Liberty and Social Constraints.” Guide the readers in their understanding.
line 260-261: “The gymnasts 260 were intended to…” By whom? Use active voice.
line 262-263: “first in the 263 German Confederation, then in the German Reich, in the Weimar Republic and finally in the Federal Republic of Germany.” You cover a long stretch of time in this statement. I’m not clear, for example, how the gymnasts have contributed “civic, democratic engagement” in the FRG.
line 271: “The latter did not want the gymnasts at 271 their side at all.” What evidence can you cite here to support this statement?
line 275-277: “They adapted to the political system or submitted to the constraints and controls exercised by the state and the party.” Good! Here we get a hint at the significance of the developments in the Turner movement for the broader topic of the paper.
line 277: “Gymnastics and sports organisations that still stood in political or ideological opposition to the regime were banned and persecuted…” I’m a bit cautious here. Was there really far-spread, active opposition among German gymnasts, clubs, and federations against the Nazis?
line 284: “and before that a thug.” What did Bloch mean here?
line 289: “SED regime in the GDR” (a) The first time you use an abbreviation, spell out the full name or title and introduce the abbreviation in parentheses. After that use only the abbreviation. (b) Here you seem to open a new “can of worms” that calls for a longer discussion. Without the space to provide such a discussion, consider cutting this part.
line 295-296: “In numerous writings, which he distinguished from political geography…” It is not clear what this means. Technically, it means that de Coubertin thought of his writings as apart from political geography. I don’t think that is what you mean.
line 304-306: “Everyone should be able to participate in sport, without discrimination on age, origin ("race") or gender….” Add a citation here to an IOC source.
line 321: “7. Re 2. Utopias…” It is not clear what “Re.2” means.
line 322: “By contrast…” It is not clear to me what the stated contrast is here.
line 328: What does “MK” mean?
line 331: Check on how to correctly format an em-dash. On a Mac, the keystroke combination is shift-option-dash. Note that there should be no space before or after the em-dash.
line 331: Consider adding an example of what such a utilitarian concern could be.
line 334-338: Please add an English translation.
line 349: “…has always…” Add support and a citation for this statement. Historians are very careful with the use of words like “always;” for historians that might mean from the beginning of human existence. Unless you are positive that the statement is true, rephrase the statement.
line 382-390: Toward the end of this section of the manuscript, I once again yearn for stronger connections between the present argument and “the question of which constraints stand in the way of realising this utopian perspective of fair and humane sport as a general human right” (lines 7-8). You get to this more in the next section, but I think it would be beneficial to guide the readers understanding more closely in the preceding sections of the manuscript. I think it would strengthen the eventual main argument, findings, and conclusion.
line 399: If possible provide translation(s) for “débrouillard.”
line 408-412: The brief mentioning of Foucault and Heinilä left me yearning for a deeper discussion of these concepts and points.
line 413: “Résumé” is a relatively unusual word to use in this context in (American-)English journals. Consider using “Conclusion.”
line 413-468: Although the authors presented a well-written and insightful manuscript, I do not think the manuscript is publishable in its current form. For me, the “Résumé” section reveals major issues with the manuscript. Several of the conclusions presented in this section do not seem to be related to the previous discussion. See for example the following statement,
There are many obstacles to the realisation of this right, first and foremost, poverty and malnutrition. Without a minimum material standard of living, i.e. sufficient food, clothing, housing, work and income, it is barely possible for individuals to exercise their human right to sport and exercise. (lines 422-425)
Although this seems to be a correct statement and I personally agree with it, this conclusion does not appear to be drawn from the previous discussions (e.g., Olympism, Turner movement, physical emancipation).
Many of the concluding statements and arguments in this section could have been made without the preceding information the authors provided. For another example, see this statement:
Finally, cultural and religious factors also stand in the way of realising the human right to free sport. If girls and women are prevented from practising sports by strict dress codes or gender-role stereotypes, such measures also restrict the human right to sport. (lines 448-45)
To me, this conclusion is not based on the discussion and information in the manuscript, as the authors previously said little, if any, about gender related constraints, for example in the Olympic and Turner movements.
In addition, in my opinion, the authors should discuss more what contributions the current work made to existing scholarship on topics like physical education and sport as a human right, for example. In other words, what do we know now that we didn’t know before? Note that there is no return to the sprinkling of connections to the work of Elias, Foucault, and Heinilä, for example.
Author Response
Thank you for your valuable comments and suggestions on the revision of the manuscript. The editors have given us a chance to revise the manuscript, which we much appreciate.
I will of course do my best to consider the points you make regarding the revision. I will especially revise my “Résumé”, which is more than just a conclusion, including further ideas and consequences which are not analyzed in detail in the main text.
Your comments will certainly be reflected in detail in the revised manuscript.
Reviewer 3 Report
My recommendations are the following:
I recommend that the title mention that it is mainly Germany.
In the abstract to mention the most relevant conclusion of the study.
This study is a plea for physical movement, I recommend adding many bibliographic sources in all sections.
I recommend mentioning the limitations and strengths.
Lines 86-89 is not a clear idea, I recommend detailing
Author Response
Thank you for your valuable comments and suggestions on the revision of the manuscript.
I will of course do my best to consider your various points in the revision.
I will change the title as follows: Physical Education and Sport between Human Rights, Duties, and Obligations - with Particular Reference to the German Development.
The abstract will be revised and the most relevant conclusions mentioned.
Bibliographic sources will also be added, as well as the limitations and strengths of the paper.
Lines 86-89 will be clarified.
Round 2
Reviewer 3 Report
I recommend that the conclusions section be mentioned separately from limitations and strengths so that the readers may clearly understand the contribution of your work
Author Response
Thank you for your recommendation. I will follow.